# Human IRP1 Translocates to the Nucleus in a Cell-Specific and Iron-Dependent Manner

**DOI:** 10.3390/ijms231810740

**Published:** 2022-09-15

**Authors:** Wen Gu, Carine Fillebeen, Kostas Pantopoulos

**Affiliations:** 1Lady Davis Institute for Medical Research, Jewish General Hospital, Montreal, QC H3T 1E2, Canada; 2Department of Medicine, McGill University, Montreal, QC H4A 3J1, Canada

**Keywords:** iron metabolism, iron responsive element (IRE), iron–sulfur protein, translation regulation, nucleus

## Abstract

Iron regulatory protein 1 (IRP1) is a bifunctional protein with mutually exclusive RNA-binding or enzymatic activities that depend on the presence of a 4Fe-4S cluster. While IRP1 is a well-established cytosolic protein, work in a Drosophila model suggested that it may also exhibit nuclear localization. Herein, we addressed whether mammalian IRP1 can likewise translocate to the nucleus. We utilized primary cells and tissues from wild type and *Irp1*^−/−^ mice, as well as human cell lines and tissue biopsy sections. IRP1 subcellular localization was analyzed by Western blotting, immunofluorescence and immunohistochemistry. We did not detect presence of nuclear IRP1 in wild type mouse embryonic fibroblasts (MEFs), primary hepatocytes or whole mouse liver. However, we observed IRP1-positive nuclei in human liver but not ovary sections. Biochemical fractionation studies revealed presence of IRP1 in the nucleus of human Huh7 and HepG2 hepatoma cells, but not HeLa cervical cancer cells. Importantly, nuclear IRP1 was only evident in iron-replete cells and disappeared following pharmacological iron chelation. These data provide the first experimental evidence for nuclear IRP1 expression in mammals, which appears to be species- and cell-specific. Furthermore, they suggest that the nuclear translocation of IRP1 is mediated by an iron-dependent mechanism.

## 1. Introduction

IRP1 is a cytosolic post-transcriptional regulator of cellular iron metabolism [1,2]. Under iron deficiency, it exhibits RNA-binding activity, which results in the stabilization of transferrin receptor 1 (TfR1) mRNA and the translational arrest of the mRNAs encoding ferritin and ferroportin. These homeostatic responses promote iron uptake by TfR1 and inhibit iron storage within ferritin and efflux via ferroportin. Following iron acquisition, IRP1 assembles a 4Fe-4S cluster that renders it to cytosolic aconitase. The concomitant loss of the RNA-binding activity of IRP1 triggers destabilization of TfR1 mRNA and translational de-repression of ferritin and ferroportin mRNAs, aiming to prevent iron overload. Thus, IRP1 controls cellular iron metabolism and exhibits two mutually exclusive activities, which depend on transition from apo- to holo-IRP1 by an iron–sulfur cluster switch.

Using a Drosophila model, Huynh et al. [3] reported that holo-IRP1 physically interacts with glycogen branching enzyme (GBE1), involved in glycogen synthesis, and mitoNEET, a mitochondrial membrane protein involved in Fe-S repair mechanisms. Moreover, they showed that the interaction promotes translocation of holo-IRP1 from cytosol to the nucleus. Nuclear expression of both Drosophila-specific IRP1A and IRP1B isoforms was documented in cells from the body fat and the prothoracic gland. The capacity of human IRP1 and GBE1 to form a complex was previously reported in protein–protein interaction databases and validated by Huynh et al. in Drosophila S2 cells [3]. However, it remains unclear whether endogenous IRP1 can translocate to the nucleus in mammalian cells. Herein, we addressed this by biochemical and histological approaches using cells and tissues from wild type and *Irp1*^−/−^ mice, as well as human cell lines and tissue biopsies.

## 2. Results and Discussion

In an initial experiment, immortalized MEFs from wild type and *Irp1*^−/−^ mice were left untreated or iron-loaded with ferric ammonium citrate (FAC) treatment. Subcellular localization of IRP1 was analyzed by immunofluorescence using confocal microscopy, or by Western blotting following biochemical fractionation. Confocal microscopy of FAC-treated cells showed punctate staining of IRP1 throughout the cytosol and in perinuclear areas but not within nuclei of wild type MEFs (Figure 1A). Similar results were obtained with untreated MEFs (not shown). The lack of IRP1-specific nuclear staining was verified by inspecting confocal Z-stack images; contrary to the cytoplasmic IRP1 signal, the intensity of the weak nuclear signal did not change in the different planes (Appendix A). The IRP1-enriched puncta in the cytosol very likely represent Golgi and ER membranes [4]. As expected, IRP1 was undetectable in MEFs from *Irp1*^−/−^ mice. Western blot analysis corroborated these findings by demonstrating expression of IRP1 exclusively in the cytoplasmic but not in the nuclear fraction, even after FAC treatment (Figure 1B). The glycolytic enzyme glyceraldehyde 3-phosphate dehydrogenase (GAPDH) and histone H3 were used as cytosolic and nuclear markers, respectively.

Mammalian GBE1 is robustly expressed in the liver, the organ with the highest glycogen content [5,6]. Thus, we utilized primary hepatocytes from wild type and *Irp1*^−/−^ mice for IRP1 localization studies. The cells were subjected to biochemical fractionation, either following FAC treatment or not. IRP1 was only detectable in the GAPDH-expressing cytoplasm of wild type hepatocytes (Figure 2A). As expected, FAC triggered a strong induction of ferritin, a protein that stores and detoxifies excess iron. Ferritin mRNA translation is negatively regulated by IRP1 and IRP2, an IRP1 homologue that does not assemble a 4Fe-4S cluster but is rather subjected to iron-dependent degradation [2]. Thus, ferritin regulation in *Irp1*^−/−^ MEFs is apparently mediated by IRP2.

The identified interaction of IRP1 with GBE1 in the Drosophila model supports a function of GBE1 (together with mitoNEET) as IRP1 regulator. Interestingly, IRP1 and GBE1 were also reported to interact with glycogen synthase [3]. These findings provide potential links between IRP1 and glycogen metabolism and raise the intriguing possibility that IRP1 may control glycogen synthesis [7]. To address this, we measured hepatic glycogen stores in wild type and *Irp1*^−/−^ mice. However, there was no significant difference among the genotypes (Figure 2B), indicating that IRP1 does not affect glycogen levels, at least in the liver.

The predominantly cytosolic localization of IRP1 in immortalized MEFs and primary hepatocytes, without any detectable nuclear fraction, is consistent with previous microscopy data obtained with cell lines, such as human astrocytoma SW1088 cells [4], as well as murine NIH3T3 [4] and B6 [8] fibroblasts. It is also in line with the lack of IRE-binding activity in nuclear extracts from human cervical cancer HeLa cells [8] and the biochemical detection of IRP1 in cytosolic and membrane fractions of human hepatoma HepG2 and embryonic kidney HEK293 cells [9]; nuclear fractions were not evaluated in this study. However, considering that in the Drosophila model nuclear IRP1 was tissue-specific [3], it can be argued that localization experiments using mammalian cell lines or even primary cells may be limiting and thus nuclear IRP1 could escape detection. To address this, we analyzed IRP1 expression by immunohistochemistry in liver sections of wild type and *Irp1*^−/−^ mice. In the mouse liver, the vast majority (>90%) of IRP1 contains a 4Fe-4S cluster and exhibits aconitase activity [10]. Thus, if nuclear translocation of IRP1 were relevant in this tissue, this should be evident in liver sections. The data in Figure 2C do not show any apparent nuclear IRP1 expression in the wild type mouse liver; the specificity of this experiment is shown by the neat control (wild type liver lacking primary IRP1 antibody) and the *Irp1*^−/−^ control liver, which indicates some non-specific staining (red arrowheads).

Next, we analyzed IRP1 expression in human liver and ovary biopsies by immunohistochemistry. The data in Figure 3A show primarily cytoplasmic staining of IRP1 throughout all tissue sections. However, several apparently IRP1-positive nuclei were noted within hepatocytes in the liver specimens (arrows in left and middle panels). The specificity of this signal could not be strictly assessed due to lack of IRP1-deficient controls. However, the absence of nuclear IRP1 staining in the ovary tissue (Figure 3A, right panel) seems to exclude the possibility for a false positive result in the liver. Thus, the immunohistochemical analysis of human biopsies suggests that a fraction of IRP1 is expressed in the hepatocellular nucleus.

We utilized human Huh7 and HepG2 hepatoma cells, as well as HeLa cells to validate this unexpected finding. The cells were left untreated, iron-loaded by treatment with FAC, or iron-depleted by treatment with the iron chelator desferroxamine (DFO). Subcellular localization of IRP1 was analyzed by Western blotting. The data in Figure 3B–D show cytosolic IRP1 expression in all cell types. However, a fraction of IRP1 is also expressed in the nucleus of Huh7 and HepG2, but not HeLa cells. Importantly, nuclear IRP1 translocation appears to increase in response to FAC treatment and is completely abrogated by iron chelation with DFO. As expected, the iron manipulations triggered coordinate regulation of ferritin and TfR1 expression in all cell types.

The biochemical data in human Huh7 and HepG2 hepatoma cells corroborate the immunohistochemical data in the human liver and demonstrate for the first-time nuclear expression of IRP1 in a mammalian setting, by analogy to the Drosophila model [3]. Our findings are in line with the detection of active TCA cycle enzymes in mammalian cell nuclei [11,12,13]. Moreover, we provide evidence that the mechanism for nuclear translocation of IRP1 depends on iron, which is likewise consistent with the findings in Drosophila. Thus, the absence of nuclear IRP1 expression in DFO-treated cells suggests that only holo-IRP1 retains the capacity to translocate to the nucleus under iron-replete conditions, while apo-IRP1 remains exclusively in the cytosol under iron deficiency. Mechanistic details and functional implications remain to be established. Further studies are also needed to understand the basis for the apparent species- and cell-specificity underlying nuclear expression of mammalian IRP1.

## 3. Materials and Methods

### 3.1. Mice

*Irp1*^−/−^ mice [14] and wild type littermates were housed in macrolone cages according to standard guidelines. At experimental endpoints, the animals were sacrificed by cervical dislocation. Experimental procedures were approved by the Animal Care Committee of McGill University (protocol 4966).

### 3.2. Cell Culture

Primary MEF cultures were established using embryos excised from uterus at 10–12 days of gestation, according to standard procedures [15]. The MEFs were immortalized using pBABE-neo largeTcDNA (Addgene, Watertown, MA, USA; #1780). Primary hepatocytes were prepared from adult mice using a two-step collagenase perfusion technique as described [16]. Human Huh7 and HepG2 hepatoma cells and HeLa cervical carcinoma cells were cultured in Dulbecco’s modified Eagle’s medium supplemented with 10% heat-inactivated fetal bovine serum (Wisent, St. Bruno, QC, Canada), non-essential amino acids, 100 U/mL penicillin and 100 μg/mL streptomycin. Where indicated, the cells were treated with ferric ammonium citrate (FAC; from Sigma-Aldrich, St. Louis, MI, USA) or desferroxamine (DFO; from Novartis, Dorval, QC, Canada).

### 3.3. Biochemical Fractionation and Western Blotting

Whole cell lysate was obtained by lysing cells in a buffer containing 10 mM Tris (pH 8.4), 140 mM NaCl, 1.5 mM MgCl_2_, 0.5% NP-40, and 1 mM DTT. For biochemical fractionation, the whole cell lysate was centrifugated in the cold room for 3 min at 600× *g*, to sediment nuclei and separate cytoplasmic supernatant. The nuclear pellet was rinsed and resuspended in the same lysis buffer. Subsequently, one-tenth volume of 3.3% sodium deoxycholate and 6.6% Tween-40 was added under slow vortexing, and the nuclear suspension was centrifuged in the cold room for 3 min at 600× *g*. The pellet was washed, resuspended in RIPA buffer and sonicated 3× at 20% for 10 s to yield the nuclear fraction. Protein extracts from either whole cell lysate, cytoplasmic, or nuclear fractions were resolved by SDS-PAGE on 9% and 13% gels and transferred onto nitrocellulose membranes (BioRad, Hercules, CA, USA). The blots were blocked in 10% bovine serum albumin in phosphate buffered saline (PBS) containing 0.1% Tween-20 and incubated overnight with antibodies against IRP1 [17], H3 (Cell Signaling, Danvers, MA, USA; #9715S), GAPDH (Sigma–Aldrich, St. Louis, MI, USA; #MAB374), ferritin (Cell Signaling, Danvers, MA, USA; #4393S), TfR1 (Invitrogen, Waltham, MA, USA; #13-6800). After wash with PBS, membranes were incubated with peroxidase-coupled secondary antibodies for 1 h. Immunoreactive bands were detected by enhanced chemiluminescence.

### 3.4. Confocal Microscopy

Cells were seeded to coverslips. After fixation and incubation with IRP1 (Novus Biologicals, Littleton, CO, USA; #NBP1-87677) and chicken anti-rabbit Alexa 594 (Invitrogen; #A21442) antibodies, cells were imaged using a Zeiss LSM800 confocal microscope Airyscan with Plan-Apochromat/20×, Air/0.80 NA and Zeiss Zen 3.3 software (Zeiss, Jena, Germany).

### 3.5. Immunohistochemistry

Mouse liver and human tissue samples (two different liver and one ovary, kindly provided by the Research Pathology Facility of the Jewish General Hospital) were used for immunohistochemistry [18] with an IRP1 antibody (Novus Biologicals, Littleton, CO, USA; #NBP1-87677). Informed consent was obtained from all subjects involved in the study.

### 3.6. Glycogen Assay

Liver glycogen content was quantified by using a commercial kit (Abcam, Cambridge, UK; #ab65620), according to the manufacturer’s recommendations.

## Figures and Tables

**Figure 1 ijms-23-10740-f001:**
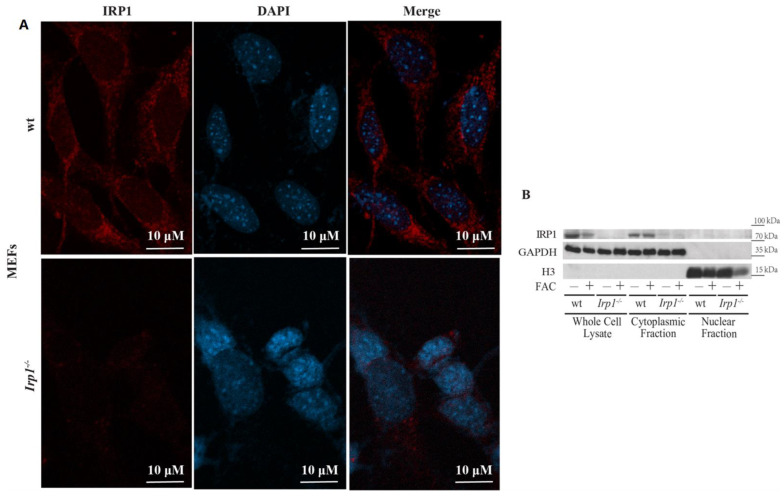
Subcellular IRP1 localization in immortalized MEFs. (**A**) Immunofluorescence of IRP1 in MEFs from wild type (wt) and *Irp1*^−/−^ mice pre-treated with 100 μg/mL FAC for 18 h; nuclei are stained with DAPI. (**B**) Western blot analysis of IRP1, GAPDH and H3 in whole cell lysates, cytoplasmic supernatants and nuclear extracts from wild type and *Irp1*^−/−^ MEFs previously treated with FAC or not.

**Figure 2 ijms-23-10740-f002:**
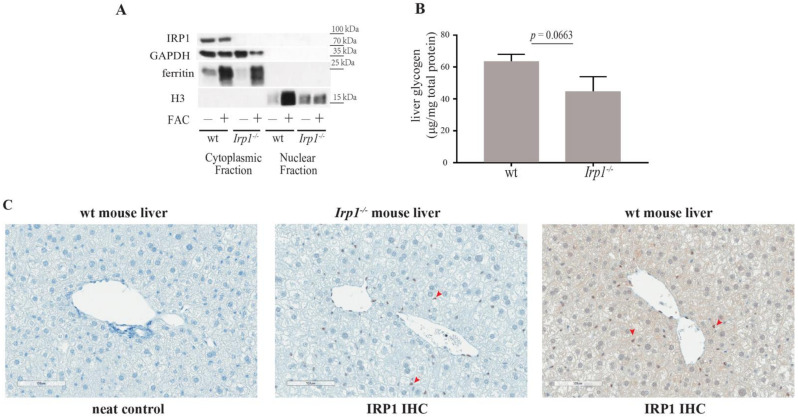
Subcellular IRP1 localization in primary murine hepatocytes and liver sections, and assessment of potential impact of IRP1 ablation in liver glycogen content. (**A**) Western blot analysis of IRP1, GAPDH, ferritin and H3 in whole cell lysates, cytoplasmic supernatants and nuclear extracts from primary wild type (wt) and *Irp1*^−/−^ hepatocytes, either previously treated with 100 μg/mL FAC for 18 h, or not. (**B**) Glycogen content in the liver of 16-week-old male wild type (*n* = 8) and *Irp1*^−/−^ (*n* = 4) mice. Statistical significance was evaluated by using the unpaired Student’s *t* test. (**C**) Immunohistochemistry (IHC) of IRP1 in the liver of wild type and *Irp1*^−/−^ mice. Red arrowheads indicate non-specific staining.

**Figure 3 ijms-23-10740-f003:**
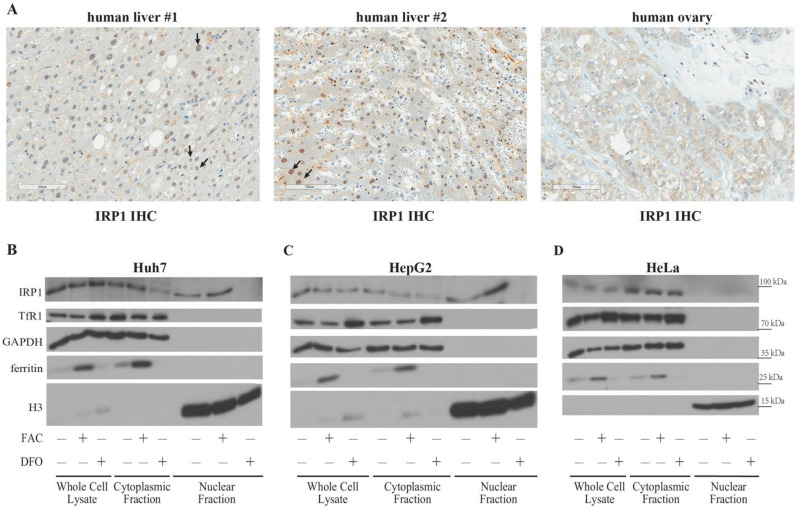
Nuclear expression of IRP1 in human liver sections and hepatoma cell lines. (**A**) Subcellular IRP1 localization in human liver and ovary biopsies. IRP1 expression was analyzed by immunohistochemistry (IHC). Arrows indicate apparent nuclear staining in the liver samples. (**B**–**D**) Western blot analysis of IRP1, TfR1, GAPDH, ferritin and H3 in whole cell lysates, cytoplasmic supernatants and nuclear extracts from Huh7 (**B**), HepG2 (**C**) and HeLa (**D**) cells previously left untreated or treated for 18 h with 100 μg/mL FAC or 100 μM DFO.

## Data Availability

All data are contained within the manuscript and the Appendix A.

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
