# Peer review of "Human IRP1 Translocates to the Nucleus in a Cell-Specific and Iron-Dependent Manner"

_ijms, 2022, doi:10.3390/ijms231810740_

Round 1

Reviewer 1 Report

This timely short communication is a welcome reaction to the discovery that Drosophila iron regulatory proteins localize to the nucleus of cells of the larval prothoracic gland and fat bodies. The authors wondered whether a similar phenomenon might take place in mammalian cells, where the iron regulatory proteins have been assumed to reside exclusively in the cytosol for many decades. The experiments shown are straightforward and performed with expected controls. The findings differ between mice and human tissues. Not finding nuclear IRP1 in some cell types and conditions in mice certainly confirms common knowledge about this protein in the field. The value of the manuscript lies in showing that similar phenomena to what was described in Drosophila are seen in human cells, where IRP1 shuttles to the nucleus in an iron dependent way in some cell types. Although the study is not extensive, it has undeniable value in setting the stage for other researchers in the field, to consider the issue. The mechanism of nuclear localization of IRP1 and its function(s) in the nucleus are new important questions for the field of iron metabolism. I recommend publication of the article as received.

Minor comment: line 42 "to repair oxidatively damaged IRP1" could be misinterpreted as a suggestion that the localization serves this function, whereas Huynh et al. suggest that the interaction with mitoNEET leads to such repair and suggest on the basis of what happens with specific mutants of IRP1 that IRP1 with the iron-sulfur cluster assembled is the dominant form detected in the nucleus.

Author Response

Please, see attachment

Reviewer 2 Report

The manuscript by Gu et al. reports that holo-IRP1, an iron sensor protein, can go inside the nucleus in the human liver cells and in Huh7 and HepG2 hepatoma cells. However, in MEFs or primary hepatocytes, or liver cells from mice, nuclear localization of IRP1 is not observed. The authors, for the first time, show a cell-type specific subcellular localization of IRP1 in human cell lines and tissue. These observations are of potential interest to the field. The manuscript is well written. Here are my comments:

1.    I am not able to understand the claim in line 36: “…transition from apo- to holo-IRP1 by an unusual iron-sulfur cluster switch.” IRP1 binding to IRE is based on iron-dependent Fe-S biogenesis. Hence, the authors should either remove “unusual” or explain why they think it is unusual.

2.    Fig. 1A resolution is not good. The authors should provide an image with better resolution.

3.    The quality of the manuscript will be increased if the authors can provide immunostaining data showing that DFO or FAC treatment affects the nuclear localization of IRP1 in Huh7 and/or HepG2 cells.

4.    In the glycogen assay, it is not clear what statistical test was used to measure significance. The present results show a trend toward a decrease although not a significant level. What is the p-value? The authors should make these points clear either in the materials and methods section or in the figure legend.

5.    The authors should mention the catalog number for the antibodies and kits used in this study.

6.    Line 52: “…or iron-loaded with ferric ammonium citrate (FAC).” should be or iron-loaded with ferric ammonium citrate (FAC) treatment.

Line 141-143: “Moreover, they provide evidence that the mechanism for nuclear translocation of IRP1 depends 142 on iron, which is likewise consistent with the findings in Drosophila.” Do the authors mean: Moreover we provide…?

Round 2

Reviewer 2 Report

Since the authors have made changes in the manuscript as per the suggestions, I do not have any further comments.